# Singularities in Euler Flows: Multivalued Solutions, Shockwaves, and Phase Transitions

Valentin Lychagin [1] and Mikhail Roop [1,2,*]

1   V.A. Trapeznikov Institute of Control Sciences of Russian Academy of Sciences, 65 Profsoyuznaya Str., 117997 Moscow, Russia; valentin.lychagin@uit.no
2   Faculty of Physics, Lomonosov Moscow State University, Leninskie Gory, 119991 Moscow, Russia
*   Correspondence: mihail_roop@mail.ru

**Abstract:** In this paper, we analyze various types of critical phenomena in one-dimensional gas flows described by Euler equations. We give a geometrical interpretation of thermodynamics with a special emphasis on phase transitions. We use ideas from the geometrical theory of partial differential equations (PDEs), in particular symmetries and differential constraints, to find solutions to the Euler system. Solutions obtained are multivalued and have singularities of projection to the plane of independent variables. We analyze the propagation of the shockwave front along with phase transitions.

**Keywords:** Euler equations; shockwaves; phase transitions; symmetries

## 1. Introduction

Various types of critical phenomena, such as singularities, discontinuities, wave fronts and phase transitions, have always been of interest from both mathematical [1–3] and practical [4] viewpoints. In the context of gases, discontinuous solutions to the Euler system, describing their motion, are usually treated as *shockwaves*. In the past decades, such phenomena have widely been studied (see, e.g., [5] for the case of Chaplygin gases [6,7], where the weak shocks are considered). It is also worth mentioning the works in [8,9], where the influence of turbulence on shocks and detonations is emphasized.

This paper can be seen as a natural continuation of the work in [10], where have considered the case of ideal gas flows. Here, we use the van der Waals model of gases, which is more complicated and at the same time more interesting from the singularity theory viewpoint. The van der Waals model is known to be one of the most popular in the description of phase transitions. Thus, singularities of shockwave type that can be viewed as in some sense singular solutions to the Euler system are analyzed together with singularities of purely thermodynamic nature, phase transitions. Our approach to finding and investigating such phenomena is essentially based on the geometric theory of PDEs [11–15]. Namely, we find a class of multivalued solutions to the Euler system (see also [16]), and singularities of their projection to the plane of independent variables are exactly what drives the appearance of the shockwave [17]. Similar ideas are used in a series of works [18–20], where multivalued solutions to filtration equations are obtained along with analysis of shocks. To find such solutions, we use the idea of adding a differential constraint to the original PDE in such a way that the resulting overdetermined system of PDEs is compatible [21]. The same concepts were also used by Schneider [22], who found a general solution to the Hunter–Saxton equation; LY1 [23], who considered the two-dimensional Euler system; and LY2 [24], who applied this approach to the Khokhlov–Zabolotskaya equation.

The paper is organized as follows. Section 2 presents the preliminary concepts, where we describe the necessary concepts from thermodynamics. In Section 3, we analyze a multivalued solution to Euler equations and its singularities, including shockwaves and

phase transitions. In the last section, we discuss the results. The essential computations for this paper were made with the DifferentialGeometry package [25] in Maple.

## 2. Thermodynamics

In this section, we give necessary concepts from thermodynamics. As shown below, geometrical interpretation of thermodynamic states allows one to use Arnold's ideas from the theory of Legendrian and Lagrangian singularities [1–3], which are crucial in description of phase transitions. The geometrical approach to thermodynamics was already initiated by Gibbs [26]. It was further developed, for example, by the authors of [27,28] and, more recently, by Lychagin [29]. For more detailed analysis regarding the geometrical methods in thermodynamics, we also refer to [30].

### 2.1. Legendrian and Lagrangian Manifolds

Consider the contact space $(\mathbb{R}^5, \theta)$ with coordinates $(s, e, \rho, p, T)$ standing for specific entropy, specific inner energy, density, pressure and temperature. The contact structure $\theta$ is given by

$$\theta = T^{-1}de - ds - pT^{-1}\rho^{-2}d\rho. \tag{1}$$

Then, a *thermodynamic state* is a Legendrian manifold $\widehat{L} \subset (\mathbb{R}^5, \theta)$, i.e., $\theta|_{\widehat{L}} = 0$ and $\dim \widehat{L} = 2$. From the physical viewpoint, this means that the first law of thermodynamics holds on $\widehat{L}$. Due to (1), it is natural to choose $(e, \rho)$ as coordinates on $\widehat{L}$. Then, a two-dimensional manifold $\widehat{L} \subset (\mathbb{R}^5, \theta)$ is given by

$$\widehat{L} = \left\{ s = S(e, \rho), \ T = \frac{1}{S_e}, \ p = -\rho^2 \frac{S_\rho}{S_e} \right\}, \tag{2}$$

where the function $S(e, \rho)$ specifies the dependence of the specific entropy on $e$ and $\rho$.

Note that determining a Legendrian manifold $\widehat{L}$ by means of (2) requires the knowledge of $S(e, \rho)$, while in experiments one usually obtains relations among pressure, density and temperature. Thus, we get rid of the specific entropy $s$ by means of projection $\pi: \mathbb{R}^5 \to \mathbb{R}^4$, $\pi(s, e, \rho, p, T) = (e, \rho, p, T)$ and consider an immersed Lagrangian manifold $\pi(\widehat{L}) = L \subset (\mathbb{R}^4, \Omega)$ in a symplectic space $(\mathbb{R}^4, \Omega)$, where the structure symplectic form $\Omega$ is

$$\Omega = d\theta = d(T^{-1}) \wedge de - d(pT^{-1}\rho^{-2}) \wedge d\rho.$$

Then, one can treat thermodynamic state manifolds as Lagrangian manifolds $L \subset (\mathbb{R}^4, \Omega)$, i.e., $\Omega|_L = 0$. In coordinates $(T, \rho)$, a thermodynamic Lagrangian manifold $L$ is given by two functions

$$L = \{p = P(T, \rho), \ e = E(T, \rho)\}. \tag{3}$$

Since $\Omega|_L = 0$, the functions $P(T, \rho)$ and $E(T, \rho)$ are not arbitrary, but are related by

$$[p - P(T, \rho), e - E(T, \rho)]|_L = 0, \tag{4}$$

where $[f, g]$ is the Poisson bracket of functions $f$ and $g$ on $(\mathbb{R}^4, \Omega)$ uniquely defined by the relation

$$[f, g] \Omega \wedge \Omega = df \wedge dg \wedge \Omega.$$

Equation (4) forces the following relation between $P(T, \rho)$ and $E(T, \rho)$: $(-\rho^{-2}T^{-1}P)_T = (T^{-2}E)_\rho$, and therefore the following theorem is valid:

**Theorem 1.** *The Lagrangian manifold L is given by means of the Massieu–Planck potential* $\phi(\rho, T)$

$$p = -\rho^2 T\phi_\rho, \quad e = T^2\phi_T. \tag{5}$$

**Remark 1.** *Having given the Lagrangian manifold L by means of (3), one can find the entropy function $S(e, \rho)$ solving the overdetermined system*

$$T = \frac{1}{S_e}, \; p = -\rho^2 \frac{S_\rho}{S_e}$$

*with compatibility condition (4).*

### 2.2. Riemannian Structures, Singularities, Phase Transitions

There is one more important structure arising, as shown in [29], from measurement approach to thermodynamics. Indeed, if one considers equilibrium thermodynamics as a theory of measurement of random vectors, whose components are inner energy and volume $v = \rho^{-1}$, one drives to the universal quadratic form on $(\mathbb{R}^4, \Omega)$ of signature $(2,2)$:

$$\kappa = d(T^{-1}) \cdot de - \rho^{-2} d(pT^{-1}) \cdot d\rho,$$

where $\cdot$ is the symmetric product of differential forms, and areas on $L$, where the restriction $\kappa|_L$ of $\kappa$ to $L$ is negative, are those where the variance of a random vector $(e, v = \rho^{-1})$ is positive [29,31]. Using (5), we get

$$\kappa|_L = -(2T^{-1}\phi_T + \phi_{TT})dT \cdot dT + (2\rho^{-1}\phi_\rho + \phi_{\rho\rho})d\rho \cdot d\rho, \tag{6}$$

and, taking into account (5), we conclude that the condition of positive variance is satisfied at points on $L$, where

$$e_T > 0, \quad p_\rho > 0,$$

which is known as the condition of the thermodynamic stability.

Let us now explore singularities of Lagrangian manifolds. We are interested in the singularities of their projection to the plane of intensive variables $(p, T)$, i.e., points where the form $dp \wedge dT$ degenerates. We assume that extensive variables $(e, \rho)$ may serve as global coordinates on $L$, i.e., the form $de \wedge d\rho$ is non-degenerate everywhere. The set where $dp \wedge dT = 0$ coincides with that where $2\rho^{-1}\phi_\rho + \phi_{\rho\rho} = 0$, or, equivalently, where the from $\kappa|_L$ degenerates. A manifold $L$ turns out to be divided into submanifolds $L_i$, where both $(e, \rho)$ and $(p, T)$ may serve as coordinates, or, equivalently, the form (6) is non-degenerate. Such $L_i$ are called *phases*. Additionally, those of $L_i$, where (6) is negative, are called *applicable phases*. Thus, we end up with the observation that singularities of projection of thermodynamic Lagrangian manifolds are related with the theory of phase transitions. Indeed, by a *phase transition* of the first order, we mean a jump from one applicable state to another, governed by the conservation of intensive variables $p$ and $T$ and specific Gibbs potential

$$\gamma = e - Ts + p/\rho,$$

which in terms of the Massieu–Planck potential is expressed as $\gamma = -T(\phi + \rho\phi_\rho)$ [30]. Consequently, to find the points of phase transition, one needs to solve the system

$$p = -\rho_1^2 T\phi_\rho(T, \rho_1), \quad p = -\rho_2^2 T\phi_\rho(T, \rho_2), \quad \phi(T, \rho_1) + \rho_1\phi_\rho(T, \rho_1) = \phi(T, \rho_2) + \rho_2\phi_\rho(T, \rho_2), \tag{7}$$

where $p$ and $T$ are the pressure and temperature of the phase transition and $\rho_1$ and $\rho_2$ are the densities of gas and liquid phases.

**Example 1** (Ideal gas)**.** *The simplest example of a gas is an ideal gas model. In this case, the Legendrian manifold is given by*

$$\widehat{L} = \left\{ p = R\rho T, \; e = \frac{n}{2}RT, \; s = R\ln\left(\frac{T^{n/2}}{\rho}\right) \right\}, \tag{8}$$

*where R is the universal gas constant and n is the degree of freedom. The differential quadratic form* $\kappa|_L$ *is*

$$\kappa|_L = -\frac{Rn}{2}\frac{dT^2}{T^2} - R\rho^{-2}d\rho^2.$$

It is negative definite on the entire $\widehat{L}$, and there are neither phase transitions nor singularities of projection of $\widehat{L}$ to the $p - T$ plane.

**Example 2** (van der Waals gas). *To define the Legendrian manifold for van der Waals gases, we use reduced state equations:*

$$\widehat{L} = \left\{ p = \frac{8T\rho}{3 - \rho} - 3\rho^2, \, e = \frac{4nT}{3} - 3\rho, \, s = \ln\left( T^{4n/3}(3\rho^{-1} - 1)^{8/3} \right) \right\}. \tag{9}$$

*The differential quadratic form* $\kappa|_L$ *is*

$$\kappa|_L = -\frac{4n}{3T^2} dT^2 + \frac{6(\rho^3 - 6\rho^2 - 4T + 9\rho)}{\rho^2 T(\rho - 3)^2} d\rho^2.$$

*In this case, it changes its sign; the manifold* $\widehat{L}$ *has a singularity of cusp type. The singular set of* $\widehat{L}$, *called also caustic, and the curve of phase transition are shown in Figure 1.*

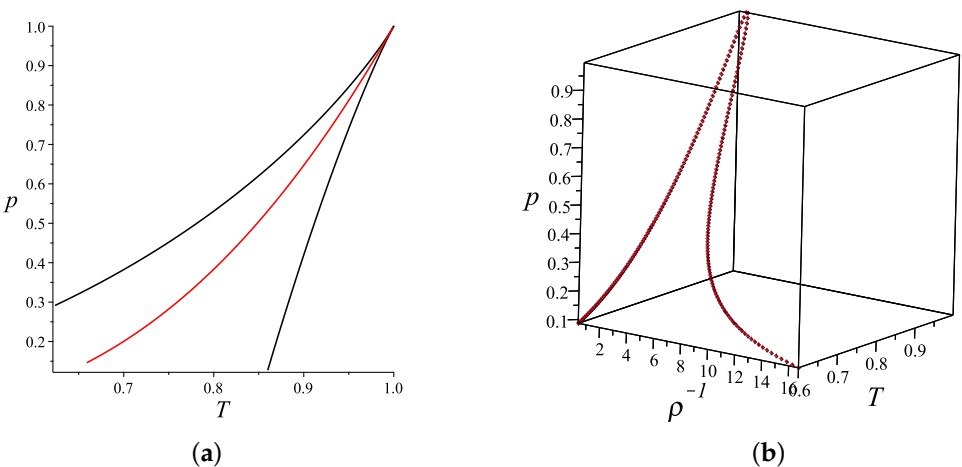

(**a**)   (**b**)

**Figure 1.** Singularities of the van der Waals Legendrian manifold: caustic (black line) and phase transition curve (red line) in coordinates $(p, T)$ (**a**); and the curve of phase transition in $(p, \rho, T)$ (**b**). Points of the phase transition curve with the same values of pressure $p$ and temperature $T$ and different values of density $\rho_2 > \rho_1$ correspond to the liquid phase and the gas phase, respectively, while points between $\rho_1$ and $\rho_2$ correspond to wet steam.

## 3. Euler Equations

In this paper, we study non-stationary, one-dimensional flows of gases, described by the following system of differential equations:

- Conservation of momentum:

$$\rho(u_t + uu_x) = -p_x \tag{10}$$

- Conservation of mass:

$$\rho_t + (\rho u)_x = 0 \tag{11}$$

- Conservation of entropy along the flow:

$$s_t + us_x = 0 \tag{12}$$

Here, $u(t, x)$ is the flow velocity, $\rho(t, x)$ is the density of the medium, and $s(t, x)$ is the specific entropy. System (10)–(12) is incomplete. It becomes complete once extended by equations of thermodynamic state (2). We are interested in *homentropic flows*, i.e., those with $s(t, x) = s_0$. On the one hand, this assumption satisfies (12) identically. On the other

hand, it allows us to express all the thermodynamic variables in terms of $\rho$. Indeed, the entropy $s$ has the following expression in terms of the Massieu–Planck potential $\phi(T, \rho)$: $s = \phi + T\phi_T$ [30]. Putting $s = s_0$, we get the equation $s_0 = \phi + T\phi_T$, which determines $T(\rho)$ uniquely, since the derivative of its right-hand side with respect to $T$ is positive due to the negativity of $\kappa|_L$. Substituting $T(\rho)$ into (3), one gets $p = p(\rho)$. Thus, we end up with the following two-component system of PDEs:

$$u_t + uu_x + A(\rho)\rho_x = 0, \quad \rho_t + (\rho u)_x = 0, \tag{13}$$

where $A(\rho) = p'(\rho)/\rho$.

We do not specify the function $A(\rho)$ yet; we do this while solving (13).

### 3.1. Finding Solutions

To find solutions to system (13), we use the idea of adding a differential constraint to (13), compatible with the original system. It is worth mentioning that a solution is an integral manifold of the Cartan distribution on (13) (see [11–13] for details). This geometrical interpretation of a solution to a PDE allows finding ones in the form of manifolds, which, in general, may not be globally given by functions. This approach gives rise to investigation of singularities in a purely geometrical manner, which is shown in this paper.

In general, finding differential constraints is not a trivial problem. However, having found ones, the problem of finding solutions is reduced to the integration of a completely integrable Cartan distribution of the resulting compatible overdetermined system. In rgw case the Cartan distribution has a solvable transversal symmetry algebra, whose dimension equals the codimension of the Cartan distribution, we are able to get explicit solutions in quadratures by applying the Lie–Bianchi theorem (for details, see [11–13]).

We look for a differential constraint compatible with (13) in the form of a quasilinear equation

$$u_x - \rho_x(\alpha(\rho)u + \beta(\rho)) = 0, \tag{14}$$

where functions $\alpha(\rho)$ and $\beta(\rho)$ are to be determined. We denote system (13) and (14) by $\mathcal{E}$.

**Theorem 2.** *System (13) and (14) is compatible if*

$$\alpha(\rho) = -\frac{1}{\rho(C_3\rho - 1)}, \quad \beta(\rho) = \frac{C_2}{\rho(C_3\rho - 1)}, \quad A(\rho) = C_1 + \frac{C_5}{\rho^3}\left(C_3 + \frac{C_7}{\rho}\right)^{C_6}, \tag{15}$$

*where $C_i$ are constants.*

The proof of Theorem 2 is more technical rather than conceptual. First, we lift system (13) and (14) to the space of 3-jets $J^3(\mathbb{R}^2)$ by applying total derivatives

$$\begin{aligned} D_t &= \partial_t + u_t\partial_u + \rho_t\partial_\rho + u_{tt}\partial_{u_t} + \rho_{tt}\partial_{\rho_t} + \dots, \\ D_x &= \partial_x + u_x\partial_u + \rho_x\partial_\rho + u_{xx}\partial_{u_x} + \rho_{xx}\partial_{\rho_x} + \dots. \end{aligned}$$

to equations of $\mathcal{E}$ the required number of times, consequently. The resulting system $\mathcal{E}_3 \subset J^3(\mathbb{R}^2)$, consisting of equations only of the third order, contains nine equations for eight variables of purely third order: $u_{ttt}, u_{xxx}, u_{txx}, u_{ttx}, \rho_{ttt}, \rho_{xxx}, \rho_{txx}$ and $\rho_{ttx}$. Eliminating them from $\mathcal{E}_3$, we get seven relations (six obtained by lifting $\mathcal{E}$ to $J^2(\mathbb{R}^2)$ plus one remaining from eliminations of third-order variables). Again, we eliminate all the variables of the second order and we get four relations of the first order. Eliminating $u_x$, $u_t$ and $\rho_t$, we end up with an expression of the form $\rho_x^3 G(\rho, u) = 0$, where $G(\rho, u)$ is a polynomial in $u$, whose coefficients are ordinary differential equations (ODEs) on $\alpha(\rho)$, $\beta(\rho)$ and $A(\rho)$, solving which we get (15). It is worth stating that these computations are algebraic and well suited for computer algebra systems.

**Remark 2.** *Using (8) and (9), one can show that the function $A(\rho) = p'(\rho)/\rho$ given in (15) corresponds to that of:*

- *ideal gas in the case of*

$$C_1 = C_3 = 0, \quad C_5 = R\left(1 + \frac{2}{n}\right)\exp\left(\frac{2s_0}{Rn}\right), \quad C_6 = -2 - \frac{2}{n}, \quad C_7 = 1;$$

- *van der Waals gas in the case of*

$$C_1 = -6, \quad C_3 = -1, \quad C_5 = 24\left(1 + \frac{2}{n}\right)\exp\left(\frac{3s_0}{4n}\right), \quad C_6 = -2 - \frac{2}{n}, \quad C_7 = 3. \tag{16}$$

*The case of ideal gases was thoroughly investigated by LR2 [10]. Here, we are interested in the case of van der Waals gases.*

Summarizing, we have a compatible overdetermined system of PDEs

$$\mathcal{E} = \{F_1 = u_t + uu_x + A(\rho)\rho_x = 0, F_2 = \rho_t + (\rho u)_x = 0, F_3 = u_x - \rho_x(\alpha(\rho)u + \beta(\rho)) = 0\} \subset J^1(\mathbb{R}^2),$$

where functions $\alpha(\rho)$, $\beta(\rho)$ and $A(\rho)$ are specified in (15). This system is a smooth manifold $\mathcal{E}$ in the space of 1-jets $J^1(\mathbb{R}^2)$ of functions on $\mathbb{R}^2$. Since dim $J^1(\mathbb{R}^2) = 8$, and $\mathcal{E}$ consists of three relations on $J^1(\mathbb{R}^2)$, dim $\mathcal{E} = 5$. The dimension of the Cartan distribution $\mathcal{C}_{\mathcal{E}}$ on $\mathcal{E}$ equals 2, therefore codim $\mathcal{C}_{\mathcal{E}} = 3$. Let us choose $(t, x, u, \rho, \rho_x)$ as internal coordinates on $\mathcal{E}$. Then, the Cartan distribution $\mathcal{C}_{\mathcal{E}}$ is generated by differential 1-forms

$$\omega_1 = du - u_x dx - u_t dt, \tag{17}$$

$$\omega_2 = d\rho - \rho_x dx - \rho_t dt, \tag{18}$$

$$\omega_3 = d\rho_x - \rho_{xx} dx - \rho_{xt} dt, \tag{19}$$

where $\rho_{xx}$, $\rho_{xt}$, $u_t$, $u_x$, $\rho_t$ are expressed due to $\mathcal{E}$ and its prolongation $\mathcal{E}_2 = \{D_t(F_1) = 0, D_t(F_2) = 0, D_t(F_3) = 0, D_x(F_1) = 0, D_x(F_2) = 0, D_x(F_3) = 0\}$:

$$\rho_{xx} = \frac{\rho_x^2\left(\rho(C_3\rho - 1)^3 A' + (C_3\rho - 1)^2 A + 3C_3(C_2 - u)^2\right)}{(C_3\rho - 1)((C_2 - u)^2 - A\rho(C_3\rho - 1)^2)}, \quad \rho_t = \frac{\rho_x(C_3\rho u + C_2 - 2u)}{1 - C_3\rho}, \tag{20}$$

$$u_x = \frac{\rho_x(C_2 - u)}{\rho(C_3\rho - 1)}, \quad u_t = -\frac{\rho_x(A\rho(C_3\rho - 1) + u(C_2 - u))}{\rho(C_3\rho - 1)}, \tag{21}$$

$$\rho_{xt} = \frac{\rho_x^2}{\rho(C_3\rho - 1)^2(A\rho(C_3\rho - 1)^2 - (C_2 - u)^2)}\Big(\rho^2(C_3\rho - 1)^3(C_3\rho u + C_2 - 2u)A' +$$
$$+ \rho A(C_3\rho - 1)^2(C_3\rho u + 3C_2 - 4u) + (C_2 - u)^2(3C_3^2\rho^2 u + 3C_3\rho(C_2 - 2u) - 2C_2 + 2u)\Big), \tag{22}$$

where $A(\rho)$ is given by (15). We look for integrals of the distribution (17)–(22), which give us an (implicit) solution to (13) and (14).

**Theorem 3.** *The distribution (17)–(22) is a completely integrable distribution with a three-dimensional Lie algebra $\mathfrak{g}$ of transversal infinitesimal symmetries generated by vector fields*

$$X_1 = t\partial_t + x\partial_x - \rho_x\partial_{\rho_x}, \quad X_2 = \partial_t, \quad X_3 = \partial_x$$

*with brackets $[X_1, X_3] = -X_3$, $[X_1, X_2] = -X_2$, $[X_2, X_3] = 0$.*

*The Lie algebra $\mathfrak{g}$ is solvable, and its sequence of derived algebras is*

$$\mathfrak{g} = \langle X_1, X_2, X_3\rangle \supset \langle X_2, X_3\rangle \supset 0.$$

Thus, the Lie–Bianchi theorem [11–13] can be applied to integrate (17)–(22).

Let us choose another basis $\langle \varkappa_1, \varkappa_2, \varkappa_3 \rangle$ in $\mathcal{C}_{\mathcal{E}}$ by the following way:

$$\begin{pmatrix} \varkappa_1 \\ \varkappa_2 \\ \varkappa_3 \end{pmatrix} = \begin{pmatrix} \omega_1(X_1) & \omega_1(X_2) & \omega_1(X_3) \\ \omega_2(X_1) & \omega_2(X_2) & \omega_2(X_3) \\ \omega_3(X_1) & \omega_3(X_2) & \omega_3(X_3) \end{pmatrix}^{-1} \begin{pmatrix} \omega_1 \\ \omega_2 \\ \omega_3 \end{pmatrix}.$$

Due to the structure of the symmetry Lie algebra $\mathfrak{g}$, the form $\varkappa_1$ is closed [11,12], and therefore locally exact, i.e., $\varkappa_1 = dQ_1$, where $Q_1 \in C^\infty(J^1)$, while restrictions $\varkappa_2|_{M_1}$ and $\varkappa_3|_{M_1}$ to the manifold $M_1 = \{Q_1 = \text{const}\}$ are closed and locally exact too. Integrating the differential 1-form $\varkappa_1$ we observe that variables $u, \rho, t, x$ can be chosen as local coordinates on $M_1$ and

$$M_1 = \left\{ \rho_x = \frac{\alpha_1 \rho^2 (C_3 \rho - 1)}{\rho A (C_3 \rho - 1)^2 - (C_2 - u)^2} \right\},$$

where $\alpha_1$ is a constant. Integrating restrictions $\varkappa_2|_{M_1}$ and $\varkappa_3|_{M_1}$, we get two more relations that give us a solution to (13) and (14) implicitly:

$$t + \alpha_2 + \frac{C_2 - u}{\alpha_1 \rho} + \frac{C_3 u}{\alpha_1} = 0, \tag{23}$$

and

$$\begin{aligned} 0 = x + \alpha_3 + \frac{1}{\alpha_1} \Bigg( &C_1 \ln \rho - C_1 C_3 \rho + \frac{C_3 u^2}{2} + \frac{u(C_2 - u)}{\rho} - C_5 \left( C_3 + \frac{C_7}{\rho} \right)^{C_6 + 1} \cdot \\ &\cdot \frac{2 \rho^2 C_3^2 - C_7^2 (C_6 + 1)(C_3 \rho (C_6 + 3) - C_6 - 2) + C_3 C_7 \rho (C_3 \rho (C_6 + 3) - 2C_6 - 2)}{(C_6 + 1)(C_6 + 2)(C_6 + 3) C_7^3 \rho^2} \Bigg), \end{aligned} \tag{24}$$

where we have already substituted $A(\rho)$ from (15), and $\alpha_2, \alpha_3$ are constants. The graph of a multivalued solution for the density is shown in Figure 2. We used substitution (16), where $C_5 = 240$, $n = 3$, together with $C_2 = 1$, $\alpha_1 = 1$, $\alpha_2 = 2$, $\alpha_3 = 1$.

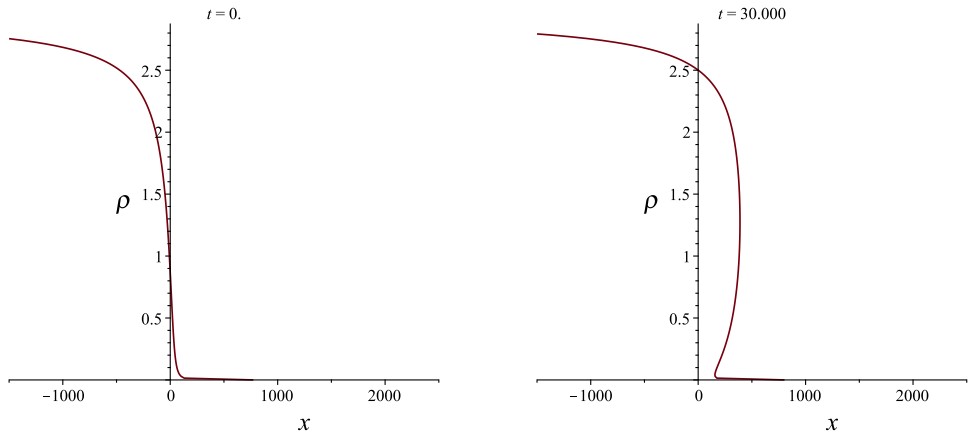

**Figure 2.** Graph of the density in case of $n = 3$ for time moments $t = 0$, $t = 30$.

### 3.2. Caustics and Shockwaves

We can see that solution given by (23) and (24) is, in general, multivalued. To figure out where the two-dimensional manifold $N$ given by (23) and (24) has singularities of projection to the plane of independent variables, one needs to find zeroes of the two-form

$dt \wedge dx$. Condition $(dt \wedge dx)|_N = 0$ gives us a curve in the plane $\mathbb{R}^2(t, x)$ called *caustic*. Choosing $\rho$ as a coordinate on the caustic, we get its equations in a parametric form:

$$
x(\rho) = -\frac{1}{2\alpha_1}\left(2C_1 \ln \rho + C_1(C_3^3\rho^3 - 4\rho^2 C_3^2 + 3C_3\rho - 2) + C_3 C_2^3 + 2\alpha_1\alpha_3\right) \pm
$$
$$
\pm \frac{C_2(C_3\rho - 1)^2}{\alpha_1\rho^2}\sqrt{C_1\rho^3 + C_5\left(C_3 + \frac{C_7}{\rho}\right)^{C_6}} - \frac{C_5\left(C_3 + \frac{C_7}{\rho}\right)^{C_6}}{2(C_6 + 2)(C_6 + 3)C_7^3\alpha_1(C_6 + 1)\rho^3} \cdot
$$
$$
\cdot\left(C_3^3(-4 + C_7^3(C_6^3 + 6C_6^2 + 11C_6 + 6) + (-2C_6 - 6)C_7)\rho^3 - \right.
$$
$$
- 2C_7((2(C_6^3 + 6C_6^2 + 11C_6 + 6))C_7^2 + (-C_6^2 - 3C_6)C_7 - C_6)C_3^2\rho^2 +
$$
$$
\left. + C_7^2(C_6 + 1)((C_6 + 3)(5C_6 + 12)C_7 - 2C_6)C_3\rho - 2C_7^3(C_6 + 4)(C_6 + 2)(C_6 + 1)\right), \tag{25}
$$

$$
t(\rho) = -\alpha_2 - \frac{C_2 C_3}{\alpha_1} \pm \frac{(C_3\rho - 1)^2}{\alpha_1\rho^2}\sqrt{C_1\rho^3 + C_5\left(C_3 + \frac{C_7}{\rho}\right)^{C_6}}. \tag{26}
$$

To construct a discontinuous solution from the multivalued one given by (23) and (24), we use the mass conservation law. Equation (11) with the velocity $u$ found from (23) in terms of $t$ and $\rho$ takes the form:

$$
\rho_t + \left(\rho\frac{\alpha_1\rho(t + \alpha_2) + C_2}{1 - C_3\rho}\right)_x = 0,
$$

and therefore the conservation law is

$$
\Theta = \rho dx - \rho\frac{\alpha_1\rho(t + \alpha_2) + C_2}{1 - C_3\rho}dt.
$$

Its restriction $\Theta|_N$ to the manifold $N$ given by (23) and (24) is a closed form, locally $\Theta|_N = dH$, and the potential $H(\rho, t)$ equals

$$
H(\rho, t) = \frac{\rho}{2\alpha_1(C_3\rho - 1)^2}\left(C_1 C_3^3\rho^3 - 4C_1 C_3^2\rho^2 + \rho\left(C_2^2 C_3^2 + (2C_2(t + \alpha_2)\alpha_1 + 5C_1)C_3 + \alpha_1^2(t + \alpha_2)^2\right) - 2C_1\right) -
$$
$$
- \frac{C_5\left(C_3 + \frac{C_7}{\rho}\right)^{C_6}}{(C_6 + 2)\alpha_1 C_7^2(C_6 + 1)\rho^2}(C_3\rho + C_7)(C_3(1 + (C_6 + 2)C_7)\rho - (C_6 + 1)C_7).
$$

The discontinuity line, or a shockwave front, is found from the system of equations

$$
H(\rho_1, t) = H(\rho_2, t), \quad x(\rho_1, t) = x(\rho_2, t),
$$

where $x(\rho, t)$ is obtained from (23) and (24) by eliminating $u$. Caustics along with the shockwave front are shown in Figure 3. Note that the picture is similar to that in the case of phase transitions.

The final result here is the expression for the time interval, within which the solution (23) and (24) is smooth.

**Theorem 4.** *The solution given by (23) and (24) is smooth and unique in the time interval* $t \in [0, t^*)$, *where*

$$
t^* = \frac{1}{\alpha_1}\left(-C_2 C_3 - \alpha_1\alpha_2 + (C_3 - 3)^2\sqrt{\frac{C_1}{27} + C_5(C_3 + 3C_7)^{C_6}}\right),
$$

*and in the case of (16), where* $C_5 = 240$, $n = 3$, *together with* $C_2 = 1$, $\alpha_1 = 1$, $\alpha_2 = 2$, $\alpha_3 = 1$ *approximately* $t^* = 12.53$.

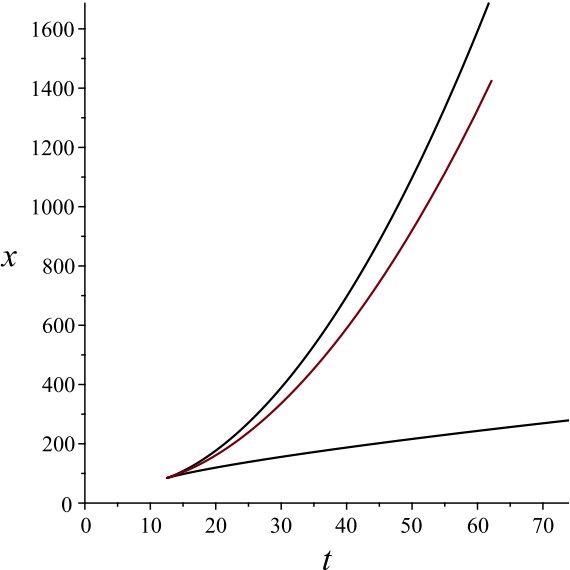

**Figure 3.** Caustic (black) and shockwave front (red) for $n = 3$.

### 3.3. Phase Transitions

Having a solution, one can remove the phase transition curve from the space of thermodynamic variables to $\mathbb{R}^2(t, x)$. Indeed, on the one hand, we have all the thermodynamic parameters as functions of $(t, x)$. On the other hand, we have conditions on phase transitions (7) in the space of thermodynamic variables. In combination, they give us a curve of phase transitions in $(t, x)$ plane. Phase transitions together with the shockwave are presented in Figure 4. We use substitution (16), where $C_5 = 240$, $n = 3$, together with $C_2 = 1$, $\alpha_1 = 1$, $\alpha_2 = 2$, $\alpha_3 = 1$.

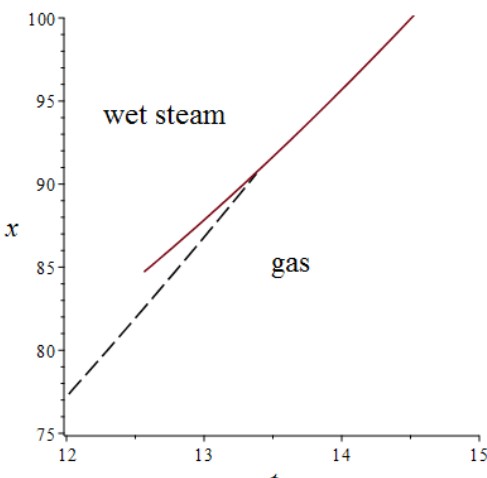

**Figure 4.** Phase transition curve (dash line) and shockwave front (red line).

## 4. Discussion

In the present work, we analyze critical phenomena in gas flows of purely thermodynamic nature, which are phase transitions and shockwaves arising from singularities of solutions to the Euler system. To obtain such solutions, we use a differential constraint compatible with the original system. In this work, it is found in a purely computational way, and how to get it in a more constructive way seems interesting. One possible way to find such constraints is using differential invariants. Then, constraints can be found constructively by solving quotient PDEs (see [32] for details), which was successfully realized by Schneider [22]. We hope to make use of this method in future research. The

analysis of phase transitions shows that sometimes shockwaves can be accompanied with phase transitions, which is shown in Figure 4, since the phase transition curve intersects the shockwave front, and on the one side of the discontinuity curve we observe a pure gas phase, while on the other side we can see a wet steam.

**Author Contributions:** Conceptualization, V.L.; Formal analysis, M.R.; Investigation, V.L. and M.R.; and Writing—original draft, M.R. All authors have read and agreed to the published version of the manuscript.

**Funding:** This work was partially supported by the Russian Foundation for Basic Research (project 18-29-10013) and by the Foundation for the Advancement of Theoretical Physics and Mathematics "BASIS" (project 19-7-1-13-3).

**Conflicts of Interest:** The authors declare no conflict of interest.

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
