# Peer review of "Singularities in Euler Flows: Multivalued Solutions, Shockwaves, and Phase Transitions"

_symmetry, doi:10.3390/sym13010054_

Round 1

Reviewer 1 Report

No comments

Author Response

Thank you for your attention to the paper.

Reviewer 2 Report

Abstract (line 3) - Please mention Partial Differential Equations (PDEs) before the abbreviation

Page 2 line 39-41 - Please break down the long sentence into a couple of sentences which will give more accurate and clear message. Last sentence (ends with "... refer to [30]") gives the impression that as if it isn't completed.

Page 6 line 138 - What are the 8 third-order variables? Please give them all instead of mentioning 3 variables only.

Page 6 line 142 - Before shortening the ODE, it is good to mention the long name first.

Page 8 line 183 - The sentence starts with "Let us ...". It would be good to use formal language. 

Reviewer 3 Report

See attached file
